# Nerve Ultrasound of Peripheral Nerves in Patients Treated with Immune Checkpoint Inhibitors

**DOI:** 10.3390/medicina59061003

**Published:** 2023-05-23

**Authors:** Katharina Kneer, Jan-Hendrik Stahl, Cornelius Kronlage, Paula Bombach, Mirjam Renovanz, Natalie Winter, Alexander Grimm

**Affiliations:** 1Department of Neurology and Epileptology, University Hospital Tübingen, Eberhard Karls University Tübingen, 72076 Tübingen, Germany; 2Hertie Institute for Clinical Brain Research, Eberhard Karls University Tübingen, 72076 Tübingen, Germany; 3Department of Neurology and Interdisciplinary Neuro-Oncology, University Hospital Tübingen, Eberhard Karls University Tübingen, 72076 Tübingen, Germany; 4Center of Neuro-Oncology, Comprehensive Cancer Center Tübingen-Stuttgart, University Hospital Tübingen, Eberhard Karls University Tübingen, 72076 Tübingen, Germany

**Keywords:** immune checkpoint inhibitors, polyneuropathy, nerve ultrasound, UPSS

## Abstract

*Background and Objectives*: Immune checkpoint inhibitors (ICIs) have enriched tumor therapy, improving overall survival. Immunotherapy adverse events (irAEs) occur in up to 50% of patients and also affect the peripheral nervous system. The exact pathomechanism is unclear; however, an autoimmune process is implicated. Thus, the clinical evaluation of irAEs in the peripheral nervous system is still demanding. We retrospectively analyzed nerve ultrasound (NU) data of polyneuropathies (PNPs) secondary to checkpoint inhibitors. *Materials and Methods*: NU data of patients with PNP symptoms secondary to ICI therapy were retrospectively analyzed using the Ultrasound Pattern Sum Score (UPSS) as a quantitative marker. Our findings were compared with a propensity score match analysis (1:1 ratio) to NU findings in patients with chronic inflammatory demyelinating polyneuropathy (CIDP) and chemotherapy-associated PNP patients. *Results*: In total, 10 patients were included (4 female, mean age 66 ± 10.5, IQR 60–77), where NU was performed in 80%. The UPSS obtained ranged from 0 to 5 (mean 2 ± 1.6, IQR 1–2.5). The morphological changes seen in the NUs resembled sonographic changes seen in chemotherapy-associated PNP (n = 10, mean UPSS 1 ± 1, IQR 0–2) with little to no nerve swelling. In contrast, CIDP patients had a significantly higher UPSS (n = 10, mean UPSS 11 ± 4, IQR 8–13, *p* < 0.0001). *Conclusions*: Although an autoimmune process is hypothesized to cause peripheral neurological irAEs, NU showed no increased swelling as seen in CIDP. The nerve swelling observed was mild and comparable to ultrasound findings seen in chemotherapy-associated PNP.

## 1. Introduction

The development and use of immune checkpoint inhibitors (ICIs) revolutionized modern tumor therapy [1]. By blocking key receptors involved in the T-cell-mediated immune response, such as programmed cell death protein-1 (PD-1), its ligand (PDL-1) or cytotoxic T-lymphocyte-associated protein 4 (CTLA4), an innate immune system-mediated tumor therapy is achieved. To date, ICI therapy can be used in nine different tumor entities, and often the mechanism of action of an ICI is limited to the microenvironment of the tumor tissue [2].

As the PD1, PDL-1 and CTLA4 receptors are coexpressed in non-tumor related tissue, a systemic down-regulation due to ICI therapy is further associated with the development of side effects, called immune-related adverse events (irAEs) [3,4]. A majority of irAEs occur during the treatment induction phase, yet individual case reports describe patients suffering from irAE symptoms up to 12 months after discontinuation of ICI therapy [5,6,7].

Approximately 50% of all patients experience side effects due to ICI therapy, and neurological irAEs (nirAEs) account for 1–5% of all registered cases. Yet, numerous studies indicate that ICI therapy is overall better tolerated than “conventional” chemotherapy [8,9,10,11,12,13].

To date, there is no single predicting factor that can identify patients at risk of developing adverse effects or their severity. It could be observed that the use of a combined ICI therapy regimen, as well as the total dose administered and treatment setting can increase the overall risk [8,11,12,13,14].

Neurological symptoms experienced by patients vary widely, but can be subdivided by central and peripheral complications.

The most common peripheral neurological irAEs seen is myasthenia gravis (MG), which is often accompanied by myositis, Guillain–Barre syndrome (GBS), axonal polyneuropathies (PNPs), chronic inflammatory demyelinating polyneuropathy (CIDP) or polyradiculitis [12,15,16,17,18,19].

As symptoms surrounding nirAEs may mimic complications due to patients’ comorbidities or side effects due to prior therapy, many cases remain unrecognized, and therefore, the obscurity rate is thought to be higher than the rate of cases recognized.

Besides nerve conduction studies (NCSs), the use of nerve ultrasound (NU) may help to identify a nerve’s morphology, and thus may enable us to further stratify and identify the cause of polyneuropathy. Ultrasound may characterize the echogenicity of a nerve but also the enlargement of the cross-sectional area (CSA). Nerve swellings can be focally, regionally or diffusely restricted and may be unifocal, multifocal or generalized. The Ultrasound Pattern Sum Score (UPSS) is a score quantifying nerve enlargements found in the peripheral sensorimotor nerves (UPS A), the C5/6 nerve roots and vagus nerve (UPS B), and the sensory nerves (UPS C), with 0 being the lowest and 22 the highest score possible. Particularly in immune-mediated neuropathies, increased UPSSs have been found in most cases (up to 90%) [20]. Further, a UPSS that is lower than 3 points likely excludes an autoimmune inflammatory process [20].

The aim of the study was to identify and describe the nerve morphology of nirAE patients suffering from polyneuropathy symptoms secondary to ICI therapy. Further, our aim was to identify changes that may help further distinguish between other differential diagnoses.

## 2. Materials and Methods

### 2.1. Inclusion Criteria

All patients diagnosed and treated due to polyneuropathy symptoms secondary to ICI therapy from January 2018 to November 2022 were retrospectively included.

An analysis was conducted based on age, sex, malignancy and cerebrospinal fluid laboratory baseline markers, including total cell count (in cells per µL), total protein (in mg/dL), and NCS and nerve ultrasound (NU) results (for baseline characteristics, see Table 1).

The clinical examination of each patient was standardized and includes the assessment of gait; muscle strength; sensation, including pinprick/soft touch and vibration; reflexes; and the strength of both upper and lower extremities.

### 2.2. Nerve Conduction Studies and Nerve Ultrasound

Nerve conduction studies and nerve sonography were conducted by trained neurologists from the University Clinic Tuebingen. (Ultrasound: Canon Aplio 900i, 24 MHz, Cadwell Neurography System).

The NCSs included the most involved side of the motor median, ulnar, tibial and fibular nerves in addition to F-waves; for the median and ulnar nerves, they were stimulated from the wrist to Erb’s point if tolerated at a predefined temperature. We performed sensory NCSs for the sural and the ulnar nerve.

The findings from the NCSs were categorized as demyelinating or axonal according to Preston and Shapiro.

CIDP diagnosis was classified as CIDP and possible CIDP according to the EAN criteria 2021 [21,22].

In patients presenting with symmetric polyneuropathy, nerve ultrasound was performed on the right side. In asymmetric polyneuropathy symptoms, ultrasound was performed at the most affected site, such as, for example, the right arm and left leg. The UPSS is a sum score that includes measurements of the nerve cross-sectional area (CSA in mm^2^) of the median, ulnar, tibial and fibular nerves (UPS A, maximum total of 16 points); the diameter of the nerve roots of C5 and C6 as well as the CSA of the vagus nerve (UPS B, maximum total of 3 points); and the sural nerve (UPS C, maximum total of 1 point) on predefined sites [20]. The assessment of nerve enlargement is based on several cross-sectional data of healthy adults [23]. In a nerve enlargement >150%, 2 points are given, and in a nerve enlargement >100% but less than 150%, 1 point is given.

### 2.3. Control Cohort

A propensity score match analysis was conducted (1:1). The matching criteria were based on the age and sex of the patient cohort. The comparator group included patients with chronic inflammatory demyelinating polyneuropathy and patients with axonal polyneuropathy secondary to chemotherapy. (Table 2 and Table 3).

### 2.4. Exclusion Criteria

Patients with other possible causes of PNP, such as diabetes mellitus, prior tumor therapy with chemotoxic agents strongly associated with PNP (for example, cisplatin or carboplatin), hyper- or hypovitaminosis, copper deficiency, and meningeosis carcinomatosa, have been excluded.

### 2.5. Statistical Significance

Statistical significance was assessed by using a one-sided *t*-test to evaluate if nirAE patients present with an increased UPSS compared to those with chemotherapy-associated polyneuropathy and CIDP. Statistical significance was defined as *p* < 0.05.

### 2.6. Ethics

This study was approved by the local ethics committee (Ethics Committee of the Medical Faculty of the Eberhard Karls University and of the University Hospital Tübingen, project number 099/2022BO2, approved on 16 February 2022). Written consent forms were signed by each patient prior to ward admission, allowing for the collection of data for scientific purposes in retrospective studies.

## 3. Results

### 3.1. Baseline Data

In total, we identified 16 cases, of which 6 patients were excluded according to the exclusion criteria. In all, the 10 included patients’ NCS findings were consistent with axonal polyneuropathy. Patients’ baseline characteristics, including mean age, malignancy, ICI agent used, mean time between ICI treatment initiation and beginning of symptoms, and ultrasound and NSC scores, are summarized in Table 1.

In total, 60% of all cases received ICI therapy for malignant melanoma, whereas one patient received ICI therapy for small-cell lung carcinoma (10%), two for urothelial carcinoma (20%) and one for esthesioneuroblastoma (10%). A total of two patients received atezolizumab monotherapy (20%), two were treated with pembrolizumab (20%), two with nivolumab (20%) and four with a combination therapy of ipilimumab and nivolumab (40%).

All our patients showed clinical signs of polyneuropathy: nine out of ten patients experienced sensory disturbances, varying from hypesthesia to dysesthesia and anesthesia. Symptoms affected the upper and lower extremities equally. In all ten patients, pallhypesthesia and areflexia were found.

Four out of ten patients experienced muscle weakness, and in two patients’ paresis not only affected the muscles of the lower extremities, but also of the upper extremities. In all patients, the distal musculature was more affected than the proximal musculature. Furthermore, a symmetrical distribution of motor and sensory symptoms was seen in all patients.

All patients were offered analgetic therapy with neuropathic agents, such as gabapentin, pregabalin, amitriptyline, lamotrigine or lacosamide. Nine out of ten patients were continuously treated with neuropathic analgesia. All patients reported persistent neuropathic pain retrospectively.

Nerve ultrasound was conducted in 8 out of 10 cases (80%). The nerve morphology identified via ultrasound was slim, with no or little signs of hypoechogenicity. The UPSS obtained varied from 0 to 5 points (mean 2 ± 1.6, IQR 1–2.5). In 80% of patients’ cerebrospinal fluid (CSF) was obtained, and in all cases pleocytosis (>5 cells/µL) and elevated protein levels (>45 mg/dL) were seen. All CSF probes were investigated by a trained neuropathologist, with none of them showing signs of meningeosis carcinomatosa.

Further, all probes were tested for common bacteria and viruses that cause CNS infections, such as the herpes simplex virus, Epstein–Barr virus, varicella zoster virus, HHV-6, borreliosis, syphilis, Strep. Pneumoniae and meningococcus. None of the probes were shown to be positive on further testing. Additionally, none of the probes were positive for common antineuronal antibodies as a possible differential diagnosis for the symptoms. In total, 50% of our patients received intravenous immunoglobulins (IVIG), and 20% received corticosteroids (CS).

### 3.2. Comparator Group

The comparator group included 10 patients with chemotherapy-associated PNP (40% female, mean age 60 ± 9.1 years, mean UPSS 1 ± 1, IQR 0–2; see Table 2) and 10 CIDP patients (20% female, mean age 67 ± 14.5 years, mean UPSS 11.8 ± 4.7, IQR 6–20) (Table 3).

Compared to the ICI-related axonal PNP patients, CIDP patients showed a significantly higher UPSS (*p* < 0.0001); no statistical significance was observed when comparing UPSS to chemotherapy-related PNP (*p* = 0.09) (Table 3).

## 4. Discussion

Polyneuropathy experienced by cancer patients due to chemo- or ICI therapy side effects negatively impacts the overall quality of life due to pain, increased imbalance, frequent falls and decreased mobility.

Neurological irAEs include polyneuropathies with different NCS findings that are demyelinating, which is comparable to CIDP, or axonal-like with toxic etiologies.

The pathomechanism underlying peripheral nirAEs remains unknown. Studies have described demyelization due to a down-regulation of ICI target receptors expressed on non-tumor-associated tissue, which may be a possible explanation for the development of CIDP secondary to ICI therapy [24].

The NCS findings in all of our patients showed predominant axonal damage. None of our patients fulfilled the electrophysiological or sonographic EAN-2021 criteria for a definite or atypical CIDP.

To date, there is a lack of defining factors identifying patients at risk for the development of peripheral nirAEs. However, melanoma patients especially seem to be at risk of developing irAEs, possibly due to shared epitopes on axons and the melanoma tissue itself causing an autoimmune reaction [19,24].

The treatment of nirAEs varies from ICI discontinuation to oral or intravenous corticosteroids (CS), immune absorption by plasma separation, or intravenous immunoglobulin therapy (IVIG), depending on the severity of symptoms [3,8].

Several studies and case reports show the effectiveness of these agents in patients with Guillain–Barre syndrome (GBS) or CIDP secondary to ICI therapy with good clinical outcomes [17,18].

However, it is unclear whether an autoimmune inflammatory process underlies all irAEs, including the development of axonal polyneuropathies.

On one hand, all of our patients showed elevated cell counts and elevated protein levels in CSF investigations with no competing cause, such as meningeosis carcinomatosa, a positive result for the presence of antineural antibodies or CNS infections (Table 1).

On the other hand, nerve sonography performed on our patients showed an inhomogenic pattern.

For example, in GBS patients, nerve root and vagus nerve swelling were predominant, whereas distal nerve segments were not predominantly swollen.

In our patient collective, 60% of cases showed nerve swelling, which was inhomogenic and did not solely affect the peripheral sensorimotor nerves (UPS A), but also affected the C5/6 nerve roots, vagus nerve (UPS B) and sensory nerves (UPS C). In 20% of patients no nerve swelling was observed, and the UPSS obtained varied from 0 to 5. (Table 1, Figure 1).

Nerve ultrasound in CIDP patients may reveal prominent single-fascicle swelling or increased intraneural echogenicity. The current data suggest that increased nerve echogenicity is associated with nerve structure remodeling, which is secondary to chronic inflammatory processes. Further, nerve swelling may affect all nerve segments, which was also found in our CIDP cohort. Compared to our CIDP patient collective, the nerve swelling observed in our nirAE patients was slight and not statistically significant, with *p* > 0.05. (Table 2, Figure 1). Although increased nerve swelling and, therefore, an increased UPSS is observed in patients with autoimmune-mediated polyneuropathies, there are exemptions to this as seen in sarcoidosis or vasculitis.

In contrast, nerve swelling and a change in nerve morphology correlated to the changes seen in axonal polyneuropathies secondary to chemotherapy.

Compared to chemotherapy-associated axonal polyneuropathies, nerve swelling in nirAE patients was more subtle (*p* < 0.0001). In most cases, no structural remodeling of the nerve was observed.

Further, we observed the treatment response to IVIG and CS in our irAE patient collective.

Subjectively, most patients described an improvement in symptoms. Yet, no improvement was observed in the NCSs or NU (Table 1).

The current literature describes an inhomogenous response towards immunomodulatory therapy.

While CIDP, GBS, myasthenia gravis, myocarditis and myositis secondary to ICI therapy seem to show a positive response towards IVIG and CS, several cases describing axonal polyneuropathies secondary to ICI therapy seem to show no positive effect [17,18,19,25].

We hypothesize that immunomodulatory therapy may be beneficial for patients presenting with CIDP secondary to ICI, whereas the treatment decision for axonal polyneuropathies should be adapted according to nerve ultrasound data.

The mean time between symptom onset, clinical presentation and treatment initiation was 5 weeks. It remains unclear if early-presenting cases treated for predominant axonal PNP and an UPSS > 3 points may be at risk to progress to a CIDP if left untreated.

On the other hand, an nirAE patient presenting with slow symptom progression, symptom persistence and predominant axonal PNP changes in the NCS with no significant nerve swelling (UPSS < 3 points) may not respond adequately to IVIG or CS therapy, as an underlying autoimmune process seems unlikely.

Further, a possible toxin-mediated axonal damage due to ICI therapy may be involved in the pathomechanism of nirAEs, which could have an impact on the poor response to immunomodulatory therapy.

Nerve biopsy, for example of the sural nerve, could give a more detailed insight into the affected nerve’s morphology and may help to manifest our understanding of nirAEs.

This study is limited by the total number of patients included, as peripheral nirAEs remain a rare complication.

Furthermore, it is unclear whether pre-damaged nerves, for example in the context of a previously known diabetic polyneuropathy, demonstrate more pronounced nerve swelling.

## 5. Conclusions

We hypothesize that the use of NU may help to identify patients that may benefit from immunomodulatory therapy. Larger prospective registry trials are required to not only confirm this hypothesis but also to clarify the underlying pathomechanism. Notably, further scoring systems can be developed based on these findings, allowing for the early identification of patients at risk and the optimization of treatment options.

## Figures and Tables

**Figure 1 medicina-59-01003-f001:**
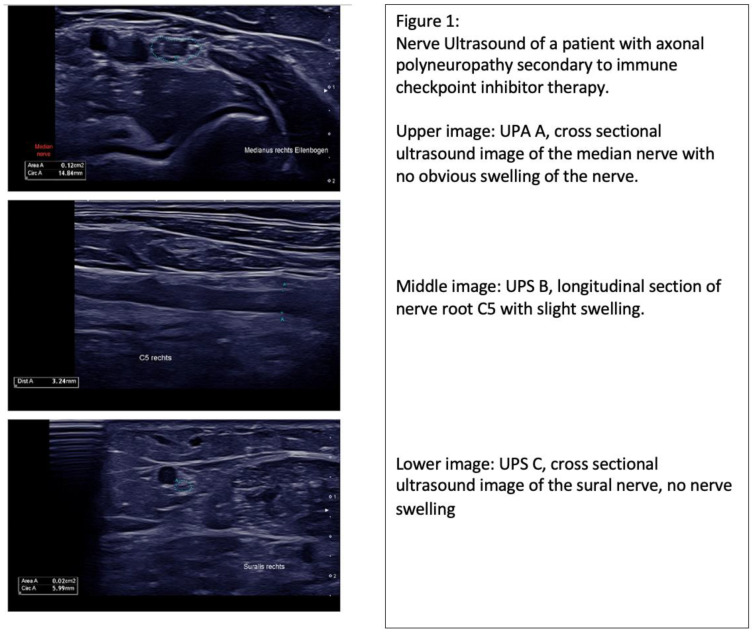
Nerve sonography data of (**top**) patient with immunotherapy-related adverse event; (**middle**) patient with chemotherapy-associated polyneuropathy; (**bottom**) patient with chronic inflammatory demyelinating polyneuropathy. Abbreviations: UPS A (Ultrasound Pattern Score A): peripheral sensorimotor nerves; UPS B (Ultrasound Pattern Score B): C5/6 nerve roots and vagus nerve; UPS C (Ultrasound Pattern Score): sensory nerves.

**Table 1 medicina-59-01003-t001:** Baseline characteristics of irAE patients.

Patient	Age	Sex	Malignancy	ICI	UPSS	NCS	LP	Treatment	Treatment Response
1	77	M	Melanoma	Ipilimumab/Nivolumab	5 UPS A 2 UPS B 2 UPS C 1	Inconclusive	N/A	N/A	N/A
2	56	M	Small-cell lung carcinoma	Atezolizumab	2 UPS A 0 UPS B 2 UPS C 0	Axonal PNP	10/µL cells, 156 mg/dL protein	IVIG	Negative
3	53	F	Melanoma	Atezolizumab	N/A	Axonal PNP	N/A	N/A	N/A
4	65	F	Melanoma	Ipilimumab/ Nivolumab	2 UPS A 1 UPS B 1 UPS C 0	Axonal PNP	19/µL cells, 100 mg/dL protein	CS	Negative
5	55	F	Melanoma	Ipilimumab/ Nivolumab	2 UPS A 2 UPS B 0 UPS C 0	Axonal PNP	26/µL cells, 1117 mg/dL protein	IVIG	Negative
6	78	M	Aesthesioneuroblastoma	Nivolumab	N/A	Axonal PNP	18/µL cells, 144 mg/dL protein	N/A	N/A
7	79	M	Melanoma	Nivolumab	0 UPS A 2 UPS B 0 UPS C 0	Axonal PNP	9/µL cells, 40 mg/dL protein	CS	Negative
8	86	M	Urothelial Ca	Pembrolizumab	3 UPS A 1 UPS B 2 UPS C 0	Axonal PNP	13/µL cells, 77 mg/dL protein	IVIG	Negative
9	68	M	Urothelial Ca	Pembrolizumab	2 UPS A 2 UPS B 0 UPS C 0	Axonal PNP	9/µL cells, 32 mg/dL protein	IVIG	Negative
10	58	F	Melanoma	Ipilimumab/ Nivolumab	0 UPS A 0 UPS B 0 UPS C 0	Axonal PNP	22/µL cells, 72 mg/dL protein	IVIG	N/A

Abbreviations: ICI: immune-checkpoint inhibitor, NCS: nerve conduction study, LP: lumbar puncture, M: male; F: female; PNP: polyneuropathy; N/A: not available; UPSS: Ultrasound Pattern Sum Score; UPS A: peripheral sensorimotor nerves; UPS B: C5/6 nerve roots and vagus nerve; UPS C: sensory nerves; IVIG: intravenous immunoglobulins; CS: corticosteroids.

**Table 2 medicina-59-01003-t002:** Baseline characteristics: polyneuropathy secondary to chemotherapy.

Patient	Sex	Age	UPSS
1	M	71	0
2	M	64	1
3	M	81	2
4	F	49	0
5	F	60	0
6	M	70	0
7	M	68	0
8	F	78	2
9	M	69	2
10	F	57	0

Abbreviations: M: male; F: female; UPSS: Ultrasound Pattern Sum Score.

**Table 3 medicina-59-01003-t003:** Baseline characteristics: patients with chronic inflammatory demyelinating polyneuropathy CIDP.

Patient	Sex	Age	UPSS
1	M	41	13 UPS A 7 UPS B 3 UPS C 3
2	M	86	17 UPS A 13 UPS B 3 UPS C 1
3	F	62	20 UPS A 13 UPS B 3 UPS C 4
4	M	84	11 UPS A 8 UPS B 1 UPS C 2
5	M	65	6 UPS A 4 UPS B 2 UPS C 0
6	M	69	15 UPS A 11 UPS B 2 UPS C 2
7	M	52	8 UPS A 6 UPS B 2 UPS C 0
8	F	71	6 UPS A 3 UPS B 2 UPS C 1
9	F	75	7 UPS A 3 UPS B 3 UPS C 1
10	F	45	15 UPS A 12 UPS B 2 UPS C 1

Abbreviations: M: male; F: female; UPSS: Ultrasound Pattern Sum Score.

## Data Availability

All data generated or analyzed during this study are included in this article. Further enquiries can be directed to the corresponding author.

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
