# Peer review of "Nerve Ultrasound of Peripheral Nerves in Patients Treated with Immune Checkpoint Inhibitors"

_medicina, 2023, doi:10.3390/medicina59061003_

Round 1

Reviewer 1 Report

The study aimed to identify and describe the nerve morphology in neurological immune-related adverse effects (nirAE) patients suffering from polyneuropathy secondary to immune checkpoint inhibitors therapy. The study has high significance in the field however the manuscript needs to be better presented. The manuscript needs to be further revised based on the following recommendations:

Overall recommendations:

·       I request the authors use professional English editing software and revise the sentence structures throughout the manuscript.

·       Please have the manuscript proofread by a native English speaker.

·       Please use square brackets ( “[ ]” ) for citing references within the text, and the numbering should be before the end of a sentence.

·       Please define the abbreviations before using them.

·       Table legends: Change “m” & “f” to “M” & “F”, respectively

     Specific recommendations:

Abstract:

·        Define NU, NSC, CIPD, and irAE before use

·        Lines 20-22: Please combine them as they seem to have redundant information

·        Line 23: Change the word “quantitively” to “quantitative”

·        Lines 26-28: Can be better structured

·        Line 26: Why NU was only performed in 80%?

Introduction:

·        Define PD-1, PDL-1, and CTLA1

·        Overall good information but difficult to understand due to poor sentence structure

·        Lines 66-67: The sentence needs to be further explained

·        Use consistent abbreviation for NCS

·        Line 76: Needs Reference

·        Please revise the aim statement for better clarity 

Methods:

·        Were the authors who performed data collection and analysis blinded from the patient condition?

·        How was the UPSS score categorized (0, 1, or 2)? What was the threshold used? And justification for the threshold?

·        What was the location of ultrasound imaging? Please describe in detail for reproducibility.

·        What was the experience of the ultrasound operator?

·        What was the p-value selected for determining significance?

·        Mention that the study was approved by Ethical Review Board

Results:

·        Line 151: Define IVIG and CS before using it

Discussion:

·        Line 204: The mentioned “Figure 2” is missing in the manuscript 

Conclusions:

·        Please add content to this section 

Data Availability:

·        Line 256 mentions that all collected data are included in this article and its supplementary files. But no supplementary files are attached. 

Informed Consent:

·        Even though the study is retrospective and anonymous, informed consent from subjects is required for data collection. The subjects must also be informed that their data will be used for research publication. Please provide more details on why the guidelines recommends otherwise.

Author Response

Dear Reviewer, we thank you for carefully reviewing the manuscript and your helpful advice. We addressed your issues point by point and hopefully improved the manuscript.

Reviewer:         I request the authors use professional English editing software and revise the sentence structures throughout the manuscript.

Authors:           We apologize for this inconvenience. This manuscript has been extensively revised by the author and all co-authors. Additionally, this manuscript has not only been re-written using a professional English editing software but has been as well proofread by a native English speaker.  

Reviewer:         Please have the manuscript proofread by a native English speaker.

Authors:           A native English speaker revised the manuscript.

Reviewer:         Please use square brackets ( “[ ]” ) for citing references within the text, and the numbering should be before the end of a sentence.

Authors:           Changes were made throughout the manuscript.

Reviewer:         Please define the abbreviations before using them.

Authors:           We thank the reviewer for pointing this out, all abbreviations were defined before using them

Reviewer:         Table legends: Change “m” & “f” to “M” & “F”, respectively

Authors:           Single letters used as an abbreviation have been written in capital letters.

Reviewer:         Define NU, NSC, CIPD, and irAE before use

Authors:           Changes were made accordingly.

Reviewer:         Lines 20-22: Please combine them as they seem to have redundant information

Authors:           Changes were made accordingly.

Reviewer:         Line 23: Change the word “quantitively” to “quantitative”

Authors:           Changes were made accordingly.

Reviewer:         Lines 26-28: Can be better structured

Authors:           The abstract has been revised and changes were made accordingly.

Reviewer:         Line 26: Why NU was only performed in 80%?

Authors:           Due to the rare occurrence of neurological irAE and partially also a poor understanding of this disease nerve ultrasound has not been included as an additional “gold standard” tool of electrophysiological work up of patients presenting before 2019. Since 2019 and the establishment of a neuro-oncological department, nerve ultrasound has been added to the routine investigation of every patient receiving ICI therapy.

Reviewer:         Define PD-1, PDL-1, and CTLA1

Authors:           We added the information in the manuscript (Line 4).

Reviewer:         Overall good information but difficult to understand due to poor sentence structure

Authors:           The sentence syntax has been revised. The manuscript has been further proofread by a native English speaker 

Reviewer:         Lines 66-67: The sentence needs to be further explained

Authors:           We thank the reviewer for this comment and hope that the additional information will lead to a better understanding.  

Reviewer:         Use consistent abbreviation for NCS

Authors:           We thank the reviewer for addressing this point. Changes were made accordingly.

Reviewer:         Line 76: Needs Reference

Authors:           The reference was added.

Reviewer:         Please revise the aim statement for better clarity 

Authors:           The changes were made accordingly.

Reviewer:         Were the authors who performed data collection and analysis blinded from the patient condition?

Authors:           The authors were not blinded to the patient condition.

Reviewer:         How was the UPSS score categorized (0, 1, or 2)? What was the threshold used? And justification for the threshold?

Authors:           The UPSS is a sum score that included measurements of nerve cross-sectional area (CSA in mm2) of the median, ulnar, tibial and fibular nerves (UPSA maximum total of 16 points), the diameter of the nerve roots of C5 and C6 as well as the CSA of the vagus nerve (UPS B maximum total of 3 points) and the sural nerve (UPS C, maximum total of 1 point). The assessment of nerve enlargement is based on several, cross sectional data of healthy adults. Grimm A et al 2018 (DOI 10.1016/j.clinph.2018.03.036) updated these “norm” values. In a nerve enlargement of > 150% 2 points, and in a nerve enlargement >100% but less than 150% 1 point is given.

Reviewer:         What was the location of ultrasound imaging? Please describe in detail for reproducibility.

Authors:           CSA of the median and ulnar nerve are measured on 4 sites including the upper arm (ulnar nerve approximately 2cm above the deltoid tuberosity, for the median nerve approximately 1-2 cm above the medial tuberosity), elbow, underarm and wrist. CSA of the tibial nerve is measured at the popliteal fossa and med. Malleolus; CSA of the fibular nerve is assessed at the popliteal fossa, further the CSA of the superficial fibular nerve is assessed. The CSA of the sural nerve is measured at the lateral malleolus and Vagus’ CSA is assessed close to the Carotid artery.
Cut- off values have been pre-defined, for example by Grimm A et al 2018 (DOI 10.1007/s13311-018-0609-4) or Kramer et al 2021 (doi: 10.3390/diagnostics11020211)

Reviewer:         What was the experience of the ultrasound operator?

Authors:           The ultrasound operators all had a different level of experience when conducting nerve ultrasound or nerve conduction studies. However, supervision was given by experiences ultrasound operators with a certified additional designation for nerve ultrasound, given by the German Association of Ultrasound Diagnostics (DEGUM). All data that has been collected was double checked by the supervisors.

Reviewer:         What was the p-value selected for determining significance?

Authors:           Statistical significance was defined by a p-value <0.05. This information has been added to the manuscript.

Reviewer:         Mention that the study was approved by Ethical Review Board

Authors:           We thank the reviewer for this comment.

Reviewer:         Line 151: Define IVIG and CS before using it

Authors:           The abbreviations were defined in the introduction section.

Reviewer:         Line 256 mentions that all collected data are included in this article and its supplementary files. But no supplementary files are attached. 

Authors:           We thank the reviewer for pointing out this issue. Changes were made accordingly and this sentence has been rephrased.

Reviewer:         Even though the study is retrospective and anonymous, informed consent from subjects is required for data collection. The subjects must also be informed that their data will be used for research publication. Please provide more details on why the guidelines recommends otherwise.

Authors:           Every patient admitted to the neurological ward must sign a written consent form allowing the use of data for research purposes.  

Reviewer:         Please add content to this section 

Authors:           The conclusion section has been added to the manuscript.

Reviewer 2 Report

The authors assessed patients with iatrogenic polyneuropathy by nerve ultrasound. They compared the patients treated with immune checkpoint inhibitors with CIDP and chemotherapy induced polyneuropathy. The authors found no particular nerve swelling in the patients that presented axonal Neuropathies.
The paper confirms previous results about nerve ultrasound.

Clinical presentation of the patients is not complete. Please, add data about muscle strength, pain, sensory disturbances. Add information in Methods and Results. 

Patients' history should be completed by the description of the used  symptomatic drugs. 

What about the possible association between the findings and the time since the therapy onset? Please, discuss. 

The conclusion paragraph is missing.

Author Response

Dear Reviewer, we thank you for carefully reviewing the manuscript and your helpful advice. We addressed your issues point by point and hopefully improved the manuscript.

Reviewer:         Clinical presentation of the patients is not complete. Please, add data about muscle strength, pain, sensory disturbances. Add information in Methods and Results. 

Authors:           We thank the reviewer for this comment. The information has been added to the manuscript.

Reviewer:         Patients' history should be completed by the description of the used symptomatic drugs. 

Authors:           Changes were made accordingly

Reviewer:         What about the possible association between the findings and the time since the therapy onset? Please, discuss. 

Authors:           So far, all our patients presented “within” the time frame, in which irAE symptoms may occur throughout the ICI treatment course. Our hypothesis includes these two scenarios:
1. A patient presenting early throughout the course of disease shows signs of a primarily axonal damage but is at risk for a disease progression and a clinical manifestation of a chronic demyelinating polyneuropathy. However, due to prompt treatment, this is masked and all we can assess as clinicians is solely an axonal nerve damage on nerve conduction studies.
2.  Changes observed are limited to an axonal nerve damage on nerve conduction studies.
In contrast to scenario 2, scenario 1 includes a certain nerve swelling seen in nerve ultrasound.

Reviewer:         The conclusion paragraph is missing.

Authors:           The conclusion section has been added to the manuscript

             We hope we addressed all your issues accordingly.

Round 2

Reviewer 1 Report

I thank the authors for working diligently on the comments and revising the manuscript accordingly. The manuscript is in much better shape and in acceptable condition. However, I just have few additional minor recommendations:

Abstract:

·        Define CIPD before use

·        Do not need to define NCS as it is not used in the abstract

·        Line 21: Please replace “iRAE” with “irAE”

·        Define PNS before use

·        Line 32: Please replace “CIDP” with “CIPD”

Overall:

·        Define IVIG and CS before using it

Author Response

Dear Reviewer,

we thank you for reviewing the manuscript and your helpful advice. 

Changes in the manuscript were made accordingly and highlighted in yellow.

Reviewer #1

Reviewer:         I thank the authors for working diligently on the comments and revising the manuscript accordingly. The manuscript is in much better shape and in acceptable condition. However, I just have few additional minor recommendations:

Authors:           We are very glad to hear this. We once again want to thank the review for the piece of advice given during the first round of revision. We do agree that the changes made lead to an improvement of the manuscript.

Reviewer:          Define CIPD before use

Authors:           CIPD has now been defined in the abstract, line 23

Reviewer:         Do not need to define NCS as it is not used in the abstract

Authors:           We deleted NCS from the abstract as it is not mentioned further.

Reviewer:         Line 21: Please replace “iRAE” with “irAE”

Authors:           Changes were made accordingly.

Reviewer:          Line 32: Please replace “CIDP” with “CIPD”

Authors:           The abbreviation CIDP has been corrected.  

Reviewer:         Define IVIG and CS before using it

Authors:           The terms are now defined for the first time in line 176 and 177. The definition has been added to “Table 1” Abbreviations section.

                        We hope we addressed all your issues accordingly.